# Max Planck WinDarts: High-Resolution Atmospheric Boundary Layer Measurements with the Max Planck CloudKite platform and Ground Weather Station – A Data Overview

Venecia Chávez-Medina<sup>1</sup>, Hossein Khodamoradi<sup>1</sup>, Oliver Schlenczek<sup>1</sup>, Freja Nordsiek<sup>1,2</sup>, Claudia E. Brunner<sup>1</sup>, Eberhard Bodenschatz<sup>1,3,4</sup>, and Gholamhossein Bagheri<sup>1</sup>

<sup>1</sup>Max Planck Institute for Dynamics and Self-Organization (MPI-DS), Am Faßberg 17, 37077 Göttingen, Germany <sup>2</sup>Gesellschaft für wissenschaftliche Datenverarbeitung mbH Göttingen (GWDG), Burckhardtweg 4, 37077 Göttingen, Germany

<sup>3</sup>Institute for Dynamics of Complex Systems, Georg August University of Göttingen, Friedrich-Hund-Platz 1, 37077 Göttingen, Germany

<sup>4</sup>Laboratory of Atomic and Solid State Physics and Sibley School of Mechanical and Aerospace Engineering, Cornell University, 130 Upton Hall, Ithaca NY 14853, USA

Correspondence: Gholamhossein Bagheri (gholamhossein.bagheri@ds.mpg.de)

# Abstract.

This paper presents the data set collected during the Pallas Cloud Experiment (PaCE) campaign, conducted at Pallas, Finland, between September 15 and September 28, 2022. The data set includes measurements of turbulence in the atmospheric boundary layer in both cloudy and cloud-free conditions, collected using the Max Planck CloudKite (MPCK) platform, the WinDarts, and

5 a ground weather station for near surface data. The airborne observations span altitudes from the surface up to 1510 m above ground level, with flight durations ranging from 1 hour to nearly 6 hours, while the ground weather station provides continuous measurements throughout the entire campaign. This data set provides high-resolution meteorological measurements to analyse boundary layer dynamics under different atmospheric conditions encountered during PaCE campaign. This paper describes the data collection process, the structure of the data set, and guidelines for users.

# 10 1 Introduction

15

The atmospheric boundary layer (ABL) is the lower fraction of the atmosphere in direct contact with the Earth's surface. Its depth and structure vary depending on weather conditions, latitude, terrain, and time of day, typically ranging from a few hundred meters to a couple of kilometres. Understanding the physical processes that govern its dynamics—such as turbulence, wind shear, convective structures, and entrainment—is crucial for many practical applications, including, fir example, weather prediction and aviation.

In-situ velocity, temperature, and relative humidity measurements are essential for a thorough understanding of turbulence in the ABL, as they capture real-world interactions. In particular, temperature and vertical velocity play a key role in understanding air mass behaviour, turbulence, and vertical motion, which drive atmospheric dynamics. Time series data from different regions of the ABL provide valuable insights into these processes, improving our understanding of boundary layer properties such as

depth, stability, and surface interactions. Moreover, such measurements are needed for refining numerical weather models and 20 climate simulations by better characterizing key processes like heat exchange, vertical mixing, boundary layer evolution, and convection.

Obtaining in-situ measurements of the ABL remains a significant challenge. The three most widely used techniques, namely tower-based, instrumented aircraft observations, and radiosondes, each have strengths and limitations. Tower-based measure-

- ments offer exceptional spatio-temporal resolution but are limited in altitude. Instrumented aircraft can probe the upper ABL 25 but struggle to access lower levels, and their high relative speed reduces spatial resolution. Radiosondes provide flexibility in launch locations and vertical range but are advected by atmospheric currents, preventing them from maintaining a steady altitude, which is needed to gather enough statistics at a given height.
- To address these challenges, the Max Planck WinDarts, developed by the CloudKite team, provide a novel solution. They 30 are deployed as part of the Max Planck CloudKite (MPCK) platform, which integrates a tethered balloon-kite hybrid (Helikite) along with complementary ground-based and airborne instruments. They bridge the gap between tower-based and instrumented aircraft measurements by enabling controlled profiling of the entire ABL under most atmospheric conditions. These instruments are purpose-built for profiling and characterising the turbulent dynamics of the ABL, offering cutting-edge, in-situ measurements of critical meteorological quantities, including temperature, humidity, wind speed, and pressure. Unlike radiosondes, the
- flight strategy of the WinDarts can be actively controlled, allowing for targeted observations and improved vertical profiling of 35 the ABL.

To complement the measurements obtained with the WinDarts, a ground weather station also recorded continuous meteorological quantities. These data serve as a baseline for assessing near-surface conditions and evaluating potential gradients between the surface and the altitude ranges sampled by the WinDarts. All together, the data set provides high-resolution measurements of meteorological variables, supporting researchers in studying the atmospheric boundary layer and enabling the

characterization of vertical profiles and fluxes across the surface layer, mixed layer, and entrainment zone.

The manuscript focuses on data description and not on scientific analysis, it begins with a brief introduction to the Pallas Cloud Experiment (PaCE) campaign, conducted in Pallas, Finland, in 2022. We present an overview of the campaign and its geographic location. Following, we introduce the Max Planck WinDarts, the ground weather station, and the methodology

45

40

used during the scientific flights conducted by the CloudKite team. Later, we present a detailed account of the data collected, focusing on raw and post-processed data sets. Finally, we present the file structure and give some notes on data availability, usage notes and intended end users.

This study is part of a special issue on the Pallas Cloud Experiment (PaCE), which brought together multiple observational platforms to investigate ABL processes in a sub-Arctic environment. The data set presented here complements other mea-

surements from the campaign, including remote sensing, UAV observations, and cloud microphysics. For a comprehensive 50 overview of the campaign, including its objectives, instrumentation, and experimental setup, readers are referred to Brus et al. (2025). Another set of atmospheric in-situ data measured with the Advanced Max Planck CloudKite Instrument (MPCK<sup>+</sup>) is published in Schlenczek et al. (2025) within the same special issue.

### 2 Overview of PaCE campaign

- 55 The Pallas Cloud Experiment (PaCE) was a field campaign mainly dedicated to conduct semi long-term measurements and characterise aerosols and clouds in vertical column at high resolution at the Pallas-Sodankylä Global Atmosphere Watch (GAW) Sammaltunturi station, operated by the Finnish Meteorological Institute (FMI) in northern Finland's Lapland region (Doulgeris et al., 2022; Brus et al., 2025; Gratzl et al., 2025).
- This initiative involved collaboration among various European scientific institutions, each deploying multiple mobile platforms to gather data on atmospheric properties (Brus et al., 2025). The campaign ran from September 15 to December 15, 2022, with an intensive period of measurement from September 15 to October 15, employing diverse methods to collect broad data sets. The Max Planck Institute for Dynamics and Self-Organization (MPI-DS), represented by the CloudKite team deploying the MPCK platform, operated from September 12 to September 29, 2022, during which a wide range of atmospheric conditions and phenomena were documented.
- Other participating institutions include the Finnish Meteorological Institute, the Swiss Federal Institute of Technology Lausanne, the University of Hertfordshire, the Karlsruhe Institute of Technology, and the Vienna University of Technology.

The measuring site is located at 68.0231° N and 24.1636° E, in Finnish Lapland and the western shoreline of Pallasjärvi Lake, approximately 280 m above mean sea level (MSL), and 162 km north of the Arctic Circle. The site is well-suited for in-situ measurements, as it is located within a designated airspace that spans 7 km on each side and extends to an altitude of 2 km. For location details, visit https://en.ilmatieteenlaitos.fi/pallas-atmosphere-ecosystem-supersite.

### **3** Instrumentation and methodology

### 3.1 The Max Planck CloudKite (MPCK) platform

The MPCK platform is composed of two tethered helikites (an aerostat with a helium-filled balloon and a kite attached to it) that combine helium buoyancy with aerodynamic lift from a kite, enabling stable tethered flights with operational heights of up to 2 km controllable by a winch. In PaCE we used the 250 m<sup>3</sup> helikite with a 34 m<sup>3</sup> stacked on top of it to provide extra helium

- to 2 km controllable by a winch. In PaCE we used the 250 m<sup>3</sup> helikite with a 34 m<sup>3</sup> stacked on top of it to provide extra helium wind wind lift. During the campaign, two WinDarts (see subsection 3.2) and a ground weather station (see subsection 3.3) were deployed with the MPCK platform, as illustrated in figure 2, with additional photographs in figure 3 showing a view from the ground station and figure 1 showing the MPCK platform and the ground weather station. During every flight, two WinDarts were positioned along the tether of the MPCK platform. Two other instruments were deployed as part of the MPCK's payload
- during some flights: the MPCK<sup>+</sup> and the FishBox. The MPCK<sup>+</sup> is developed by researchers at the MPI-DS to gain insights into cloud microphysics and turbulence Stevens et al. (2021) and Schröder (2023). The FishBox, measuring mostly aerosol-related quantities, is developed by scientists from the Finnish Meteorological Institute.

In this configuration of the MPCK platform, the secondary Helikite was stacked above the primary one to stabilise the tether and enhance overall buoyancy and payload capacity. This tandem arrangement allowed for a net payload of approximately

 $\sim 100 \text{ kg}$  to be lifted to an altitude of 2 km above ground level. The Helikites used in the MPCK platform were the  $250 \text{ m}^3$  and

Figure 1. The Max Planck CloudKite (MPCK) platform, and the ground weather station at Pallasjärvi during PaCE 2022.

 $34 \text{ m}^3$  Desert Star models manufactured by Allsopp Helikites. These models were selected for their ability to align with the wind and maintain a stable position within 55° from vertical, ensuring functionality across a broad range of wind conditions. The 250 m<sup>3</sup> Desert Star Helikite measures approximately 9.3 m in length and width and stands about 10 m tall. Its keel extends around 9.35 m in length and varies in height between 3.5 - 4.5 m.

The winch controls the length of the main tether through a line guidance system, allowing the flight altitude to be adjusted by reeling in or out the main tether, enabling flexible flight-height strategies. The wind lift generated by the Helikite sails was sufficient to reach altitudes at which the WinDarts could sample the mixed layer, the entrainment zone or above it during the campaign.

# 3.2 WinDarts

The Max Planck WinDarts are airborne, purpose-built probes designed as part of the MPCK infrastructure to characterise turbulence in the atmospheric boundary layer (ABL) from ground level up to 2 km above ground level (AGL). Suspended from the tether of the MPCK platform, they provide high spatio-temporal resolution measurements due to their low true air speed