# Peer review of "Max Planck WinDarts: High-Resolution Atmospheric Boundary Layer Measurements with the Max Planck CloudKite platform and Ground Weather Station – A Data Overview"

_Earth System Science Data, 2025_

## Author Comment (AC1)

**Reply to Referee 1**

October 6, 2025

Dear Referee,

Thank you very much for reviewing our manuscript. We are very grateful for the extremely helpful and constructive comments. In the following, we provide point-by-point replies to the points raised in your report. We have written the original text of the reviews in blue colour and our response in black colour.

**General comments:**

This paper attempts to provide an overview of the WinDarts measurement system deployed on the Max Planck CloudKite during the PaCE field campaign. In general, this is a worthwhile effort and interesting activity. Having said that there is a lack of detail in this paper that makes it challenging to really know what is going on. Additional work is required to provide the reader with a solid basis for using these data.

**Major Comments:**

Lines 42-53: These two paragraphs are a bit odd.The paper isn't about PaCE, per se. I understand that ESSD articles shouldn't be analysis focused, but I think that the authors need to determine whether this paper is specifically about the data collected during PaCE, or whether it's meant to be an overview of the WinDarts systems (in which case ESSD may not be appropriate. The subsequent text describes PaCE at a very, very high level, and primarily sends the reader to the Brus et al article (which is ok). But then why spend the real estate on PaCE at all? I would recommend rewriting these sections to truly focus on the details that are most important to the CloudKite deployment.

Thank you, we have modified lines 42 to 53, as well as section 2. We now present less information about PaCE in general while still presenting the necessary information for a self-comprehensive story.

Winds: It doesn't see as though the authors have truly calculated the 3D wind vector.They don't mention system pitch, roll, and yaw calculations at all (presumably these

can be obtained through the BNO 055 measurements?), and they only show system rel-
ative airflow in the Figure 9. This is not very helpful to those wanting to understand the
winds. It would also prove challenging to use for fluxes, as you don't know whether the
airflow is vertical or horizontal (you would need to correct for system pitch and roll to do
this correctly). This is a major shortcoming of the dataset currently. Is there a reason
that the winds and airflow angles aren't converted to a normal wind coordinate system?
Vectoflow is discussed, but no details are provided. Clearly system pitch, roll, and yaw
(along with a calibrated airspeed) would be required here. How are those obtained?

We thank the reviewer for this important comment. We fully agree that deriving the
true 3D wind vector is essential for maximizing the utility of the dataset. To address
this, we have added a description of the methodology required for motion correction
and have clarified the current limitations of the presented dataset. In particular, we
now provide additional figures that illustrate the limitations of the BNO sensors used on
the first-generation WinDarts. These sensors occasionally exhibit drift, and their over-
all performance was found to be insufficiently reliable for automatic motion correction.
While in some time periods their output may be usable, this requires careful manual
inspection, and thus could not be applied systematically across the dataset. In addition,
we show that the turbulent energy spectrum of the longitudinal velocity component is
not affected by platform motion for frequencies below 0.1 Hz (spatial scales of about 100
m). Following this reasoning, analyses can be restricted to these frequency bands, or
appropriate frequency-based filters can be applied, to reliably estimate quantities such
as turbulent energy dissipation rates even without explicit platform motion correction as
shown in our previous works (new citations added in the main text). Nevertheless, we ac-
knowledge that the lack of full motion correction is a limitation of this "Level 1" dataset.
These issues and possible mitigations are now explicitly stated in the manuscript. Fu-
ture deployments will integrate more accurate and stable inertial navigation units to
enable robust reconstruction of the full 3D wind vector across all relevant scales. We
will also consider anemometers other than 3D Pitot tubes, which in our configuration
exhibited an effective response of only about 10 Hz. (Second-generation WinDarts are
equipped with GNSS-enabled navigation units, eliminating reliance on magnetometers,
and include 3D ultrasonic sensors.)

Data quality: On line 166, it is said that "defective data were identified graphi-
cally".What does this mean? Were any quantitative measures or any formalized thresh-
olds used to evaluate where data may be bad? If so, lay out those details in this article.
If not, why not? This is another major shortcoming of the dataset and the paper at this
time. Data QC should be done in a reproducible manner, not simply by having someone
review the plots and make their own decisions on what are good or bad data.

Thank you for pointing this out. We have removed the sentence in question, as it
was ambiguous. To clarify the data quality control criteria, we have added the following
text:

"Defective data were removed from further analysis and from this manuscript according
to a set of a priori, uniformly applied quality control criteria. Data were marked for
exclusion when any of the following conditions held: (i) non-physical or out-of-range
values for the measured quantity, (ii) packet corruption, (iii) timestamp discontinuities

or non-monotonic sequences, and (iv) sensor dropouts or communication faults recorded by the logger."

Time stamping:More information on this would be very helpful (and on logging in general). Was there an onboard microprocessor used for logging? If so, did you log the computer clock values with each logging event? If not, how do you know that the GPS time logged was accurate for the time that the sensor pulled data? And to what level?

Thank you. We added the following information:

"All measurements were synchronised to GNSS-derived Coordinated Universal Time (UTC). Likewise, all times in this manuscript are presented in UTC. During September, Finland observes Eastern European Summer Time (EEST), which is UTC+3, meaning local time was 3 hours ahead of UTC during the campaign.

Each first-generation WinDart is equipped with a BeagleBone Blue (BB) single-board computer and an Arduino Mega 2560. The BB lacks a real-time clock; thus, the data-transmission timestamps ("log_time" in data files) are initially generated using Python's perf_counter() function. The synchronised timestamps ("time" in data files) for each data record are calculated using linear interpolation within a data chunk. This estimates when a specific measurement occurred between the start and end times of its data transmission. Positional time series recorded by the GPS refer to GNSS-derived UTC directly and do not undergo any further synchronisation.

The ground weather station employed the same acquisition codebase as the MPCK$^+$ (for more details, we refer the reader to Schlenczek et. al, 2025). The system clock was used to record events, and during post-processing, the corresponding timestamps were calibrated against GNSS-derived UTC timestamps using the onboard "GPS" sensor (see table 1). This was achieved by referencing all available GPS timestamps and applying a linear regression to correct for clock drift and offset. As all sensors within the system referenced the same internal clock, this method provided consistent synchronisation across all data streams. The resulting time alignment achieved sub-second accuracy, typically accurate to within a few milliseconds.

This synchronisation strategy results in consistent timestamping across all sensors within each WinDart and facilitates coordinated measurements with the ground weather station."

Figure 6: The large differences between sensors are a bit worrisome.These are likely outside of the accuracy specs of the sensors, correct? If so, how can you explain this and/or how can you justify selecting one sensor over another? More details on sensor selection for data products should be included in the manuscript. Also, if there other sensors are ever used for redundancy, that would also be good to explain.

Thank you for this remark. We recognize that it may not be obvious which sensors are heated. To clarify these points, we have improved Figure 6 by explicitly identifying the heated and reference sensors in the legends, and by adding supplementary plots that compare sensors and show the corrected relative humidity values, as explained in the newly added text below:

"Figure 6 presents the time series of pressure, temperature, relative humidity, and altitude recorded by all sensors onboard WD1-1 during flight 20220920.0750, along with the particle size concentration measured by the OPC. For pressure, temperature, and

relative humidity, a reference sensor was selected based on the highest accuracy and resolution specified by the manufacturer (see table 1): BMP for pressure, TMP for temperature, and SHT (heater off) for relative humidity. These reference measurements are shown in black in the figure. To evaluate inter-sensor consistency, we calculated the offsets of each sensor relative to the corresponding reference, expressed as $\Delta T$ and $\Delta RH$ for temperature and relative humidity, respectively. The temporal evolution of these offsets is displayed in panels zoomed around zero, and the mean offset over the entire flight is indicated in the panel labels. Temperature and relative humidity sensors exhibited the most significant offsets. For temperature, where TMP was used as the reference, SHT and BME showed mean offsets of 0.2 K and 0.4 K, respectively, both within the specified nominal accuracy (table 1). The heated SHT3 sensor displayed a positive mean offset of 3.6 K, as expected. The OPC temperature sensor, housed inside the WinDart body and OPC body itself, shows an elevated mean offset of 11.7 K. This offset was anticipated given the sensor location and instrument design. By contrast, the BMP pressure sensor exhibited a mean offset of 2.4 K, which exceeds its specified accuracy and cannot currently be explained. For relative humidity, SHT was chosen as the reference (non-heated) sensor. The SHT3 (reference heated sensor) reported systematically lower humidity, with a mean offset of –16.9 %, consistent with expectations for heated sensors. The OPC also showed a strong negative mean offset (–29.2 %). To evaluate the accuracy of the reference choice, we computed a corrected relative humidity for all sensors using the mixing ratio as a conserved quantity. Specifically, the mixing ratio was calculated from the heated sensor's (e.g. SHT3) temperature and humidity measurements, and then used together with the TMP temperature and BMP pressure to recover a corrected relative humidity[1]. For the SHT3, which we used as a reference heated sensor, this corrected value showed a mean offset of –1.5 % with respect to the reference (non-heated) SHT, which lies within the specified accuracy of the SHT sensor ($\pm$1.8 %). This confirms the suitability of the SHT (non-heated) and SHT3 (heated) as the reference for relative humidity."

**Minor Comments:**

Line 14: "fir example" should be "for example" Done
Line 16: "for developing a thorough" Changed it to: for a thorough
Lines 23-28: What about smaller, unmanned aircraft? These are used frequently for this purpose, can fly lower, and operated at lower airspeeds. You mention UAVs were used during PaCE, so clearly you are aware of using these systems for atmospheric science. Yes, this is also important to mention. We have added this new text: Uncrewed aircraft can operate at lower true airspeeds and be deployed for targeted measurements at specific altitudes and regions of interest. However, they typically have short endurance and can introduce rotor-induced aerodynamic disturbances. Moreover, they are often
* * *
[1]To compute the mixing ratio we used `metpy.calc.mixing_ratio_from_relative_humidity()` and to compute the relative humidity we used `metpy.calc.relative_humidity_from_mixing_ratio()`, both from the MetPy package (version 1.7.1).

not permitted to fly into clouds or beyond visual line of sight and, depending on the aircraft model and operating category, may be restricted from flying in precipitation or strong winds.

Section 2 header: "... the PaCE Campaign" We have changed "Campaign" to "campaign" if this was the only concern. If there are other issues, we would be grateful if the reviewer could provide further clarification.

Line 55: "conducting" Changed to: conduct

Line 56: "characterizing" Changed to: characterise

Line 56: "the vertical column" This was confusing and unnecessary. We deleted it.

Line 65: "included" – the campaign is over, correct? Yes, thank you for noticing. We changed it to: included

Line 75: 34 m3 what? We have reworded this sentence to ensure that it is unambiguous: In this configuration of the MPCK platform, a smaller Helikite (helium volume 34 $m^3$) was flown above the primary Helikite (helium volume 250 $m^3$) to stabilise the flight and increase overall buoyancy and payload capacity.

Line 76: wind wind? Thanks, we have fixed it.

Line 81: There is something wrong with how the references are presented. Fixed

Line 98: Not sure what you mean by "combined", here. Agreed. We removed the word.

Line 273: Fluxes of what? Heat? Momentum? Moisture? CO2? Aerosols? Added: fluxes of heat, momentum, moisture, and $CO_2$.

**Other author comments and modifications**

We have also corrected several typographical errors, as indicated in the marked manuscript.

---

## Author Comment (AC2)

**Reply to Referee 2**

October 6, 2025

Dear Referee,

Thank you very much for reviewing our manuscript. We are very grateful for the extremely helpful and constructive comments. In the following, we provide point-by-point replies to the points raised in your report. We have written the original text of the reviews in blue colour and our response in black colour.

Congratulations on an interesting data set that I am sure will serve many in the ABL community, particularly for those interested in low-altitude cloud microphysics. Your article already has key elements of a good data paper, and my suggestions below are targeted at improving it to increase its impact. The suggestions are broken down into three categories: Conceptual, targeting the use of specific wording or concepts that can be misconstrued; Organizational, targeting the best ordering of information for improved reading experience; and Textual, targeting typos and minor mistakes.

We thank the reviewer for the encouraging assessment and thoughtful suggestions. We have revised the manuscript accordingly, with point-by-point responses below; all changes are tracked in the marked version.

**– Conceptual –**

1 - High-resolution:
The article's title, abstract, and introduction refer to the MPCK and Wind Dart as a source of high-resolution atmospheric data. However, given the current information in the paper, it is not clear what the authors mean by it. Do you mean high-temporal-resolution because the sensors sample fast? Do you mean high-spatial-resolution in the XY plane because the system is allowed to drift, covering a "large" plane? Although not detailed in the paper how fast you can bring the system up and down, I imagine you are not using it to travel vertical ranges for a high-vertical-resolution. Am I correct? This is further complicated by using the term "high spatio-temporal resolution" (a term often used in the ABL literature in association with the vertical dimension) in Section 3.2 (line 97) when referring to a measurement that seems to be at a fixed height.

In the case you do mean "high spatio-temporal resolution", based on a "rolling atmosphere assumption" for the tower/tethered-based atmospheric measurements, I would

caution against it as it would indicate you are capturing averaged atmospheric be-haviours, which do not benefit from high spatio-temporal resolution measurements.

Given all these questions, I recommend that the authors refrain from using the term high-resolution in their work, or at the very least add a qualifier such as temporal.

Thank you for this valid point. We have removed ambiguous uses of "high-resolution" and, where relevant, explicitly refer to "high temporal resolution."

2 - Radiosondes:

In lines 23 - 28, you mention the pros/cons of radiosondes. It might be beneficial to add that because of their extensive operational range (35 vertical km), they have a varying vertical resolution, which yields a very limited number of observations in the ABL.

We added: Furthermore, due to their extensive operational range of up to 35 km, ra-diosondes have a non-uniform vertical resolution, resulting in a relatively sparse number of observations within the ABL.

3 - Entire ABL:

In section 1, line 32, you say the MPCK can profile the "entire ABL". However, all seems to indicate that although you can set the sensor height at any altitude, once that altitude is set, altitude changes are done in the scales of hours and not minutes (based on plots for figure 5). If that is the case, considering that ABL profiling is usually done by radiosondes, WxUAS, and tall towers in less than 10 minutes, using the term "entire ABL" is misleading because the different altitudes are sampled at different times and potentially at very different conditions. I recommend rewording this passage to clarify what is meant by "profiling the entire ABL" and how that differs from the majority of ABL profiling systems that are often interpreted as "instantaneous".

We agree. Some statements in this and other sections were qualitative and/or ambiguous. We have revised the text to provide quantitative and factual statements, e.g here we have modified the text to: They bridge the gap between tower-based and research-aircraft measurements by enabling controlled vertical profiling of the ABL or long time series (up to 7 hours for the first-generation WinDarts) at different heights—typically <2 km—under a wide range of atmospheric conditions.

4 - Platform motion correction:

The legend for Figure 9 indicates the data shown without any corrections for platform motion. Given the implications the motion has on the data, I believe this information should be explicitly given in the text's body (unless it is and I missed it).

Thank you, we agree this is an important point. We added:

"The velocity time series corresponds to the wind velocity vector as measured by the platform, without corrections for platform motion."

Furthermore, we have added a detailed explanation and a new figure, i.e. figure 9, to clarify how velocity corrections for platform motion can be performed, and why a dataset-wide application is not straightforward with the current data given the performance of the onboard inertial measurement system. We also clarify that large-scale variability (frequencies < 0.1 Hz; spatial scales >100 m) is expected to be minimally affected by platform motion.

**– Organizational –**

5 - Section 3.2.1:
I believe your article would read better if this section was promoted to a higher level as section 3.4, after describing all instruments in use.
Thank you, we have moved this subsection to section 4. We believe the paper now reads with a better flow.

6 - File Naming Convention:
The information in lines 118 to 125 does not make sense as part of section 3.2.1. (Wind Dart). Perhaps, it would be more appropriate to move it to section 4 (Data description) or even as part of section 5 (File Structure).
Same as the previous comment. We think it reads better now.

7 - Data availability statement:
For some reason, this statement is on page 9 instead of alongside the other statements on page 20. Additionally, it seems odd (for an open data paper) that the statement says data is available upon request (lines 141 through 143). It is even more odd that immediately following (lines 144 through 150) indicate it is available, and it is, as part of its uploaded assets. Please review this section.
Thank you for pointing this out. Our original intent was to indicate that the plotting scripts are available upon request; we have revised the section to clearly explain how the data can be openly accessed.

8 - Data Asset Names:
The uploaded data assets have the same name without indicating which one is CSV and which is NetCDF. Please change their names to include this information.
Thank you. We have now added this information to the title.

**– Textual –**

Line 14: "fir example" should be "for example". Done
Line 55: PaCE acronym seems to have already been defined in Line 48. Done
Section 3.1 Title: MPCK acronym seems to have already been defined in Line 30. Given the MPCK+ use, redefining it here could be unclear. Done

**Other author comments and modifications**

We have also corrected several typographical errors, as indicated in the marked manuscript.